# Overweight Prevalence Changes Before and After COVID-19 in Spain: The PESCA Program Longitudinal Outcomes 2018–2021

**DOI:** 10.3390/nu16233993

**Published:** 2024-11-21

**Authors:** F. Zarate-Osuna, C. Quesada-González, A. G. Zapico, M. González-Gross

**Affiliations:** 1ImFINE Research Group, Department of Health and Human Performance, Facultad de Ciencias de la Actividad Física y del Deporte-INEF, Universidad Politécnica de Madrid, 28040 Madrid, Spain; carlos.quesada@upm.es (C.Q.-G.); a.gzapico@upm.es (A.G.Z.); 2Pediatric Department, Quirónsalud Sur Hospital, 28922 Alcorcón, Spain; 3Pediatric Department, Quirónsalud Toledo Hospital, 45001 Toledo, Spain; 4Department of Applied Mathematics to Information and Communication Technologies, Universidad Politécnica de Madrid, 28040 Madrid, Spain; 5Centro de Investigación Biomédica en Red Fisiopatología de la Obesidad y la Nutrición (CIBEROBN), Institute of Health Carlos III, 28029 Madrid, Spain

**Keywords:** overweight, school, physical activity, COVID-19 impact, intervention program

## Abstract

**Background:** Overweight prevalence in Spain reached critical levels before the COVID-19 pandemic, which likely exacerbated this issue. The PESCA (Programa Escolar de Salud Cardio-vascular) program is a multicomponent school-based intervention, launched in 2018 with the aim of tackling this health problem and reducing overweight rates in youth. **Objectives:** (1) To analyze the efficacy of the PESCA program intervention on body composition, overweight prevalence, physical activity (PA)/sport practice, resting time, and screentime before COVID-19 and (2) to evaluate the impact of COVID-19 and the associated lockdown measures on these parameters in the studied sample. **Methods:** This longitudinal study included 207 children and adolescents from schools in Madrid (aged 2.82 to 15.84 years; 44.4% girls), with measurements taken at three time points: two before COVID-19 and one after its onset. Overweight prevalence, body fat percentage diagnosis, physical activity, resting time, and screentime were assessed. Cochran’s Q test and repeated-measures ANOVA were used to compare outcomes across the three assessment time points. **Results:** Overweight prevalence remained stable among children in the PESCA program before COVID-19 (17.87% to 19.81%). However, a significant increase was observed from point 2 to point 3, post-COVID-19 onset (19.81% to 26.57%). Similarly, healthy body composition significantly deteriorated from 63.16% at point 2 to 52.48% at point 3. PA/sport practice prevalence significantly increased until COVID-19 onset (80.19% to 91.22%) but declined thereafter (91.22% to 79.10% from point 2 to point 3). Although the differences were small, resting time significantly decreased post-COVID-19 onset (from 10.18 h at point 2 to 9.96 h at point 3), with no changes in the first period. Non-academic screentime showed a similar pattern: stable before COVID-19 and significantly increased after its onset (1.61 h at point 1; 1.70 h at point 2; 2.29 h at point 3). **Conclusions:** The PESCA program positively impacted PA/sport practice prevalence and may have provided some protection against overweight and related variables during the pre-COVID period. However, health authorities’ restrictions and lockdown policies during COVID-19 negatively affected the health and lifestyle variables studied, offsetting previous improvements.

## 1. Introduction

Prior to the onset of the COVID-19 pandemic, the prevalence of childhood overweight in Spain had already reached alarming levels, ranking among the highest in European Union countries [1]. Recent data from various representative studies further substantiate this concern. The PASOS-22 study documented a 33.4% prevalence of overweight (11.8% obesity) among 2892 children and adolescents aged 8 to 16, showing a 1.2% increase compared to data from the preceding decade [2,3]. Conversely, the ALADINO-23 study reported a 36.1% prevalence of overweight (15.9% obesity) in a sample of 12,678 children (aged 6–9), reflecting an 8.4% decrease in prevalence between 2011 and 2023 [4].

Obesity and overweight have posed significant health challenges over the past decades, with diet and physical activity (PA) habits recognized as major keys for prevention. The COVID-19 closure restrictions have exacerbated this issue, leading to an increase in obesogenic behaviors and overweight prevalence.

Globally, numerous studies confirm that the COVID-19 pandemic has undeniably exerted a detrimental impact on the health of children and adolescents [5,6,7]. This impact is particularly evident in the context of overweight and obesity, contributing to the deterioration of health-related conditions and behaviors, including changes in dietary patterns, PA, screentime, and rest/sleep duration [8].

PA plays a crucial role, with PA time predominantly decreasing due to the implementation of lockdowns [6,7,8]. Extended periods of school closures, mobility restrictions, the closure of public parks and playgrounds, and the suspension of extracurricular sports activities collectively contributed to a noticeable reduction in PA time among children and adolescents. Notably, the duration of school closures has shown a significant association with an increase in BMI [9,10]. In Spain, the total lockdown in 2020 lasted from 15 March to 26 April, with stringent restrictions on outdoor activities for children and adolescents persisting for an extended period. The third state of alarm ended on 9 May 2021, more than 400 days after the first began. Studies have demonstrated that engagement in PA during the pandemic predicted improvements in body composition [11]. Conversely, increased screentime and other sedentary activities were observed in children experiencing substantial BMI increases [12]. The negative impact of COVID-19 on PA levels and sedentary behaviors in Spain has been evident [13].

PESCA serves as a primary health-based intervention, with the overarching objective of improving both PA and dietary habits to effectively reduce the prevalence of overweight [14]. Recent studies continue to highlight the rising prevalence of childhood obesity and the critical role of school-based interventions, especially in the context of the COVID-19 pandemic, which has significantly impacted youth health behaviors [15,16,17]. Launched in Spain in 2018, our program operates within educational institutions covering all levels: early childhood education, primary, compulsory secondary education, and high school. Participation in the program is voluntary.

Given the pressing need to assess overweight prevalence following the onset of the pandemic and confirm the anticipated increase and its negative implications for cardiovascular health in children and adolescents, PESCA provides a unique opportunity to examine longitudinal data encompassing three measurement time points—two before and one after the onset of COVID-19—from October 2018 to June 2021. This allows for an evaluation of the effectiveness of a school-based overweight prevention and management program and likely underscores the impact of COVID-19 restrictions on these health outcomes.

Therefore, this study aims (1) to analyze the effectiveness of the PESCA program in terms of body composition, overweight prevalence, PA/sport practice, resting time, and screentime before the advent of COVID-19 and (2) to evaluate the impact of COVID-19 on all these parameters. This study’s contribution, compared to others, is to provide intra-subject longitudinal data.

## 2. Materials and Methods

A five-variable longitudinal analysis, following the PESCA protocol methodology previously described [14], was conducted from October 2018 to June 2021. The protocol involves collecting information from a background questionnaire and a clinical examination, as outlined in the following sections:Questionnaire: clinical, personal, and family history, including parents’ educational level, cardiovascular risk in the household, qualitative and quantitative dietary habits, PA, sleep, and sedentary activities of the subject.Body Mass Index (BMI).Bioelectrical Impedance Analysis (BIA): used for body composition analysis.Handgrip Strength Dynamometry (HSD): assessment of overall physical fitness.Medical Examination: focused on the cardiovascular system.

The protocol is conducted annually (once per academic year) across various educational levels: Preschool Education, Primary Education, compulsory secondary education, and high school. Clinical examinations are performed following a predetermined schedule, within approximately 60 min per classroom, using the following sequence and instrumentation:Height Measurement: Using SECA height gauges—SECA 213 (2019), SECA 206 (2018), or SECA 220 (2017)—with a precision of 0.1 cm (SECA, Hamburg, Germany).Handgrip Strength Dynamometry (HSD): Utilizing the Takei Physical Fitness Test T.K.K. 5001 GRIP A analog dynamometer (2018) with a measuring range of 0 to 100 kg (Takei, Tokyo, Japan). The subject, parallel to the trunk, holds the device and, upon command, exerts maximum grip strength for up to 10 s with the dominant hand, encouraged to exert maximum force.Cardiopulmonary Auscultation: Conducted with the student seated or lying on an examination table, including a skin examination and Tanner stage classificationSecond HSD Measurement: The better of the two readings is recorded as valid.Body Composition via Bioimpedance (BIA): Using the Tanita SC-240MA system (Tanita Europe BV, Amsterdam, The Netherlands), providing weight, BMI, and percentage of fat and water.

Comparative outcomes are described at three time points: PESCA1 (17 October 2018, to 29 January 2019), PESCA2 (14 October 2019, to 18 December 2019), and PESCA3 (7 May to 3 June 2021).

The study is a longitudinal analysis with a dual nature: it tracks evolutionary changes in the studied variables over time while also functioning as an intervention, since any medical act is considered as such. The diagnosis established through anamnesis, physical examination, and tests leads to treatment recommendations for PA and dietary habits.

The PESCA protocol adheres to the ethical standards of the Declaration of Helsinki and data protection regulations, including the General Data Protection Regulation (EU 2016/679) and Spain’s Organic Data Protection and Digital Rights Guarantee Law 3/2018. It was approved by the Jiménez-Díaz Foundation (CEIm-FJD) Ethical Committee on 9 October 2018 (record number 18/18).

### 2.1. Subjects

In the context of a broader intervention strategy, including 1400 completed protocols of children and adolescents from various educational centers in Madrid and Castilla La Mancha with the aim of reducing overweight and obesity in Spanish children and adolescents, complete longitudinal data for a 3-year follow-up were available for 207 children (92 females) of the Region of Madrid. The inclusion criterion was participation in the program on three consecutive occasions: PESCA1, PESCA2, and PESCA3. No other inclusion or exclusion criteria were applied, as participation in the protocol is voluntary.

Subjects were measured during the 2018–19 academic year (PESCA1, mean age 6.53 ± 2.54 years), the 2019–20 academic year (PESCA2), and the 2020–21 academic year (PESCA3). Overweight was identified according to the International Obesity Task Force (IOTF) BMI cut-offs to assess the impact of the PESCA program and the COVID-19 pandemic on selected health parameters. Epidemiologic sample features are shown in Table 1.

### 2.2. Statistical Analyses

IBM SPSS for Mac 25.0 (IBM Corporation, Armonk, NY, USA) was used for data analysis. Descriptive data are presented as mean (M) and standard deviation (SD) for quantitative variables, and by frequency percentage (%) and 95% confidence interval (CI) for qualitative ones. Statistical significance was set at 0.05 in all analyses. Cochran’s Q test was used to compare the prevalence of (1) overweight, (2) the percentage of children practicing sport/PA excluding physical education classes, and (3) healthy body composition diagnosed by bioimpedance over the longitudinal sequence. The McNemar test was conducted for pairwise comparisons between PESCA1 and PESCA2 (pre-COVID-19) and PESCA2 and PESCA3 (during COVID-19). Cramér’s V was used to categorize effect size, with the following criteria: <0.1 = irrelevant association; 0.1 to <0.2 = weak association; 0.2 to <0.4 = medium association; 0.4 to <0.6 = relatively strong association; 0.6 to <0.8 = strong association; and ≥0.8 = very strong association [18]. According to Blanca et al. [19], the robustness of the F-test to deviations from normality ensures that any potential non-normality in the variables does not pose a significant issue in this context. Mauchly’s test was assessed to prove sphericity, and the Greenhouse–Geisser correction was applied in case of a violation. Repeated measures ANOVA compared outcomes at the three assessment time points (PESCA1 to PESCA3) for (1) daily rest time average and (2) weekly non-academic screentime. The Bonferroni post hoc procedure was used for differences between groups. Partial Eta-Squared (ηp2) was used to assess effect size, with the following criteria: <0.06 = small; 0.06 to <0.14 = medium; and ≥0.14 = large effect. Paired *t*-tests were conducted for pairwise comparisons between PESCA1 and PESCA2, and PESCA1 and PESCA3. Cohen’s d was used to categorize effect size as small (0.2), medium (0.5), and large (0.8) [15].

## 3. Results

Table 2 and Table 3 present descriptive statistics for the five studied variables.

### 3.1. Overweight Prevalence: Table 2 and Figure 1

There were significant differences in overweight prevalence when comparing the three program time points (Q_2_ = 15.765, *p* < 0.001). When segmenting the sample by sex and age, it was observed that these differences were statistically significant in the male sex (Q_2_ = 13.579, *p* = 0.001) and in the age groups under 6 years of age (Q_2_ = 14.700, *p* = 0.001) and age between 6 and 9.99 years (Q_2_ = 7.538, *p* = 0.023)

PESCA program attendance did not result in a significant change in overweight prevalence during the first year, before COVID-19 ($\chi_{1}^{2}$ = 0.450, *p* = 0.503).

Nevertheless, after lockdowns, isolation rules and a restrictive health policy limiting personal contacts, overweight prevalence increased from PESCA2 to PESCA3 ($\chi_{1}^{2}$ = 7.042, *p* = 0.007; Vc = 0.245). Thus, overweight prevalence globally increased from PESCA1 to PESCA3 ($\chi_{1}^{2}$ = 12.042, *p* < 0.001; Vc = 0.226).

**Figure 1 nutrients-16-03993-f001:**
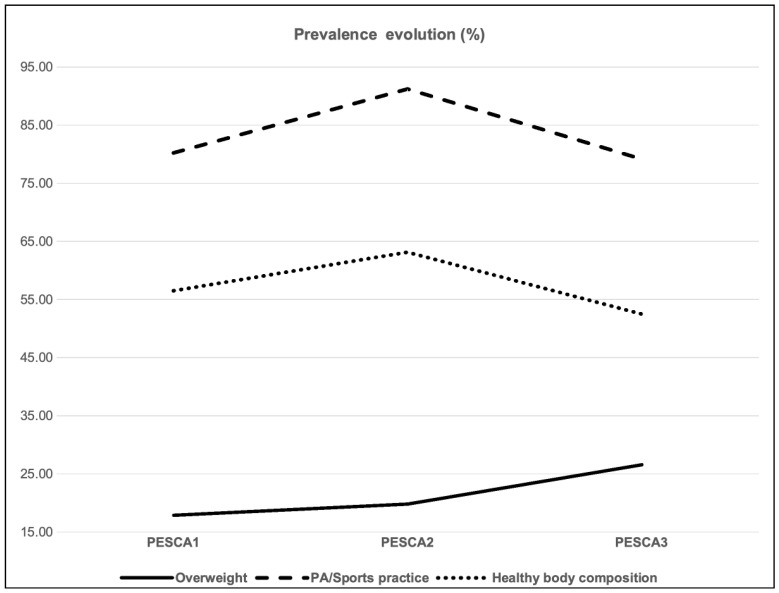
Overweight, PA/sports practice and healthy body composition prevalence evolution.

### 3.2. Healthy Body Composition: Table 2 and Figure 1

Average body fat percentage changes by sex and age. Therefore, changes in fat percentage raw data throughout the program, without any other consideration, do not allow us to have a clear diagnosis of the impact of the program or time on children. Male body fat percentage progressively increases until 10.5 years of age and then progressively decreases until at least the age of 14.5 years [20]. In addition, the increasing mean age of the longitudinal sample may be another reason for body fat percentage net changes over the three measurement points.

To solve this problem, the body fat percentage variable was categorized by sex and age in two diagnosis categories: healthy or non-healthy (including high-fat, obese, and low-fat average) body composition, according to the Tanita settings and bibliography [21,22].

Subsequently, when analyzing changes in these two diagnosis categories throughout the three measurement time points, no significant differences in healthy body composition were found (Q_2_ = 4.541, *p* = 0.103). The prevalence of healthy body composition descriptions increased from 56.47% in PESCA1 to 63.47% in PESCA2 and then, after COVID-19 onset, decreased to 52.48% in PESCA 3. Significant differences were, however, found in this last period ($\chi_{1}^{2}$ = 4.694, *p* = 0.030; Vc = 0.373).

### 3.3. Sport/PA Practice Prevalence (Excluding School Physical Education Hours): Table 2, Figure 1 and Figure 2

The prevalence of physical activity is defined as the percentage of participants who report engaging in physical activity, excluding the teaching hours of the physical education subject at school. There were significant differences in sport/PA practice prevalence throughout the three program measurement time points (Q_2_ = 15.125, *p* = 0.001). When segmenting the sample, it was observed that these differences were statistically significant in both the female (Q_2_ = 10.059, *p* = 0.007) and male sex (Q_2_ = 8.467, *p* = 0.015) and in two age groups: those between 6 and 9.99 years old (Q_2_ = 10.583, *p* = 0.005) and those under 6 years old (Q_2_ = 8.062, *p* = 0.018).

PESCA program attendance resulted in a sport/PA prevalence increase during the first year, before COVID-19 ($\chi_{1}^{2}$ = 11.605, *p* = 0.001; Vc = 0.283). After COVID-19, this prevalence decreased from PESCA2 to PESCA3 ($\chi_{1}^{2}$ = 10.023, *p* = 0.002; Vc = 0.205). Therefore, there was no global change from PESCA1 to PESCA3 for this parameter. Evolution by sex and age group is shown in Figure 2.

**Figure 2 nutrients-16-03993-f002:**
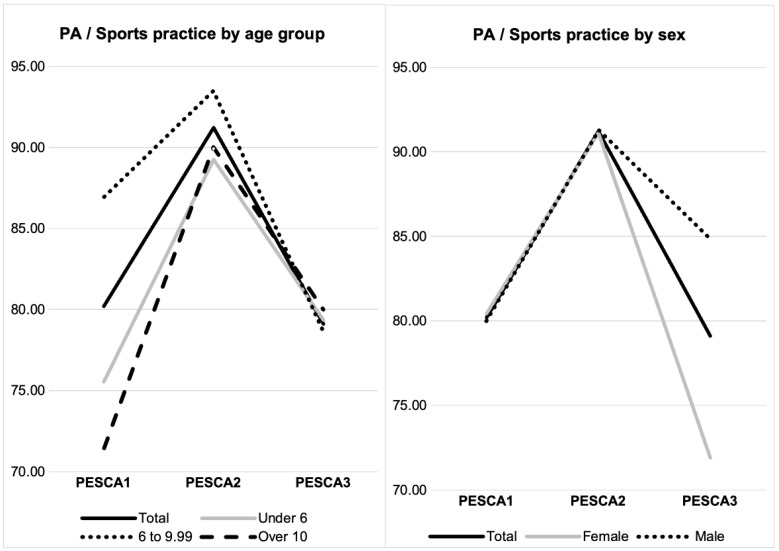
PA/sports practice evolution throughout the program.

### 3.4. Daily Rest Time Average: Table 3 and Figure 3

We define ‘resting time’ as the time elapsed, in hours, between the hour the student declares going to bed and the hour they declare waking up. As resting time may have a different change pattern over time from male to female or, most likely, from younger to older children, we have sequentially analyzed this variable considering both inter-subject factors.

**Figure 3 nutrients-16-03993-f003:**
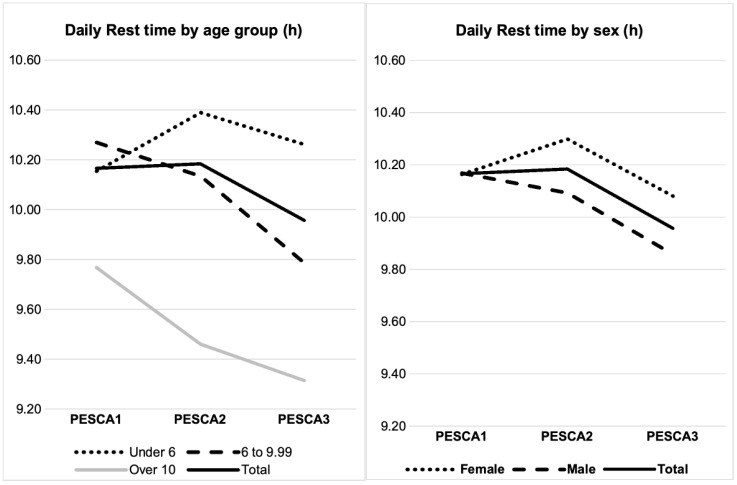
Daily rest time evolution.

When considering sex as the inter-subject factor, there were significant differences in the daily rest time average throughout the three program times points (F_2,344_ = 14.159; *p* < 0.001; ηp^2^ = 0.067). We found a significant difference in rest hours between males and females when considering the average of the three time points (*p* = 0.031). We found that, on average, males rested for less time than females. A significant interaction effect of program*sex was also evidenced (F_2,344_ = 3.784; *p* = 0.029; ηp^2^ = 0.067). The pattern of change found in rest hours over time was significantly different between males and females. In addition to differences in rest hours between males and females, the way these hours changed from one measurement point to another differed significantly between sexes. Regarding pairwise comparisons, significant differences in rest times were evidenced both between PESCA1 and PESCA3 (*p* = 0.001) and between PESCA2 and PESCA3 (*p* < 0.001). However, no significant differences were found between time points 1 and 2 (*p* > 0.05). This analysis did not change when taking age group as the inter-subject factor.

Considering age group as the inter-subject factor, significant differences in the number of rest hours were also observed across the three different measurement points (F_2,352_ = 13.558; *p* < 0.001; ηp^2^ = 0.065). We found a significant difference in rest hours between different age groups when considering the average of the three time points (*p* < 0.001). A significant interaction effect of program*age group was also observed (F_4,352_ = 12.013; *p* < 0.001; η^2^ = 0.110). The pattern of change evidenced in rest hours over time differed among different age groups so that changes in rest hours over time were not consistent across age groups. Regarding post hoc comparisons between age groups, significant differences were evidenced between all age groups (*p* < 0.05 for all comparisons).

### 3.5. Daily Non-Academic Screentime: Table 3 and Figure 4

This variable was also analyzed considering both sex and age groups as two inter-subject factors.

When considering sex as the inter-subject factor, significant differences in the daily screentime average were found throughout the three program times points (F_2,304_ = 45,121; *p* < 0.001; ηp^2^ = 0.194). Regarding the main effect of sex, we found a significant difference in screen hours between males and females when considering the average of the three time points (*p* = 0.015). On average, males spend more time in front of screens than females. However, no effect in the interaction program*sex was evidenced (F_2,304_ = 1.583; *p* = 0.210). The pattern of change found in screen hours over time did not differ significantly between males and females. Although there were differences in screen hours between males and females, the way these hours changed from one measurement point to another was similar for both sexes. Regarding pairwise comparisons, significant differences in screentime were evidenced both between time PESCA1 and PESCA3 and between PESCA2 and PESCA3 (*p* < 0.001). However, no significant differences were found between time points 1 and 2 (*p* > 0.05). This analysis did not change when taking age group as the inter-subject factor.

**Figure 4 nutrients-16-03993-f004:**
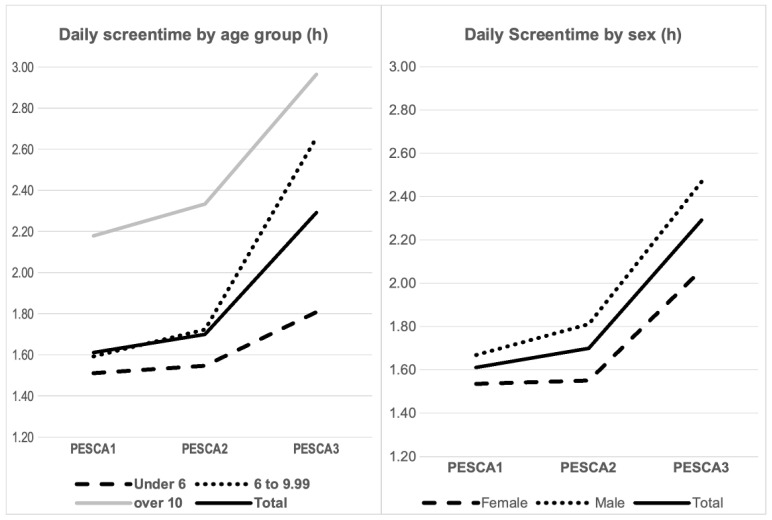
Daily screentime evolution.

Considering age group as the inter-subject factor, significant differences in the number of sport hours were also observed across the three different measurement points (F_2,312_ = 33.468; *p* < 0.001; ηp^2^ = 0.152). Significant difference in screen hours across the three measurement points were found, with a notable increase from PESCA1 to PESCA3. There was a main effect of age group: significant differences in screen hours between different age groups were found when considering the average of the three time points (*p* < 0.001). A significant interaction effect of program*age group was also observed (F_3,312_ = 7.486; *p* < 0.001; ηp^2^ = 0.074). The pattern of change found in screen hours over time differed among different age groups, so that changes in screen hours over time were not consistent across age groups. Regarding post hoc comparisons between age groups, significant differences were evidenced between all age groups (*p* < 0.05 for all comparisons).

## 4. Discussion

Consistent with the tremendous global impact of the COVID-19 pandemic on the health of children and adolescents [23], our longitudinal series reveals that overweight prevalence (including obesity) increased globally by 8.7% throughout the 43-month period measured, spanning two distinct phases: one before and another after the onset of COVID-19. Although overweight prevalence increased from 17.87 to 19.81% from PESCA1 to PESCA2, this change was not statistically significant. In contrast, the increase observed following COVID-19 onset, from 19.81% to 26.57%, was significantly higher. Even among children under six years old and those over ten and in female groups, we observed some decrease in the first period and a clear increase in the second. Although these changes were not significant, this observation could be interpreted as a minor success in the initial phase (PESCA1 to PESCA2), marking some containment in overweight prevalence, before COVID-19.

The isolation policy promoted by health authorities in Spain and the COVID-19 lockdowns could have impacted the health of children and adolescents in two distinct ways. Obesity has been shown to worsen the medical prognosis of SARS-CoV-2 infection in both child and adult patients. On the other hand, the increase in overweight and obesity prevalence due to the COVID-19 pandemic predicts a challenging scenario, with even higher rates of overweight, obesity, and cardiovascular diseases in the future [24]. In our study, the minor improvement or containment observed from PESCA1 to PESCA2 was lost, potentially influenced by the mandatory health policies adopted during the COVID-19 pandemic.

The impact of COVID-19 has been extensively documented. In the Catalonia region of Spain, changes in sleep and resting time patterns, as well as total time, were observed. A retrospective online structured survey revealed a risk of decreased total physical activity time and an increased risk of screentime [25]. Our longitudinal data support these findings within the PESCA sample during the PESCA2 to PESCA3 period, showing a significant decrease in the prevalence of PA practice and daily rest time. Medrano et al. [13] also reported in Spain, in line with our data, a deterioration in PA and screentime levels. However, children participating in the PESCA program improved the prevalence of PA practice before the pandemic, which prevented a worsening of this variable at the PESCA3 point, coinciding with the onset of COVID-19, as clearly shown in Figure 2.

When examining screentime, Cipolla et al. discovered that children with a higher BMI spent more time playing video games during the pandemic [12]. Similarly, our longitudinal data revealed a consistent and significant increase in daily screentime from PESCA2 to PESCA3, considering both globally sampled data and sex or age groups. However, there was no change in screentime hours from PESCA1 to PESCA2, showing again some kind of containment—or even protection—in children attending the PESCA program.

Regardless of its statistical significance, participation in the PESCA program implies, in our sample, an improvement in variables related to cardiovascular health in the period prior to the COVID-19 restrictions, such as PA practice prevalence or healthy body composition. Daily screentime, daily rest time, and overweight prevalence showed a different pattern: if we analyze these other three variables, there were no significant changes before COVID-19. As suggested earlier, this could be interpreted as a form of containment. However, all five variables studied worsen in the stage following the onset of the pandemic.

Regarding screentime, our data are consistent with larger series, such as the PASOS study, in which it is evident that older children and adolescents are the ones who use screens the most [2]. In our sample, all age groups increase their screentime longitudinally from PESCA2 to PESCA3, after remaining stable before COVID-19. When analyzing age and sex as inter-subject factors, we found older children and males to be the highest screen users. Therefore, in addition to promoting healthy eating and physical activity habits, it is necessary, based on the data, to make greater educational efforts regarding the time and rationalization of screen use by children and adolescents.

The COV-EAT study in Greece demonstrated that 35% of children and adolescents experienced weight gain during the first lockdown, attributed to a sudden decrease in physical activity [26]. A noteworthy study conducted in California by Dunton et al. revealed a correlation between rapid changes in eating, sports, and physical activity habits induced by COVID-19 lockdowns, along with likely permanent changes, leading to an increased risk of type 2 diabetes and cardiovascular disease [27]. In summary, the pandemic was associated with a rise in childhood obesity [28]. Similar patterns have been reported globally, indicating that these impacts on health behaviors may extend across diverse cultural and economic contexts. Therefore, urgent public health interventions are needed to mitigate the adverse effects of COVID-19 [29].

This study contributes to the knowledge in this field through two key findings. Firstly, our longitudinal data, obtained before, during, and after the COVID-19 pandemic, suggest the effectiveness of a school-based program focused on prevention, diagnosis, and health management. The program directly improved certain variables, such as the prevalence of sports practice and healthy body composition. It also played a containment and/or protective role in other areas, including the prevalence of overweight, screen hours, and sleep hours. Secondly, in line with other current studies, it highlights how COVID-19 public health policies have led to a clear regression in every analyzed variable. However, the dual clinical and research nature of the PESCA program means that, despite including more than 1400 protocols until June 2021, the longitudinal size of our sample may not yet be large enough to draw more robust and definitive conclusions. In addition, it is important to note that subgroup comparisons by age and sex may be limited by sample size and variations in adherence to the PESCA program during confinement, potentially affecting the robustness of these findings in spite of including age as a covariate in the ANOVA to improve result accuracy.

## 5. Conclusions

Given these limitations, the following conclusions are presented: (1) The PESCA program had a positive impact on PA practice prevalence and may have provided some protection in other cardiovascular health-linked variables in the pre-COVID-19 period. (2) The health authorities’ restrictions and lockdown policies had a negative effect on health and lifestyle variables studied in our sample, offsetting previous improvements in children and adolescents attending the program.

In light of the available evidence and the longitudinally presented data, there is a need for thorough reflections on restrictive health laws and policies. A comprehensive assessment of the risk–benefit balance is essential for health authorities to effectively address potential pandemic situations in the future.

## Figures and Tables

**Table 1 nutrients-16-03993-t001:** Epidemiological sample features.

	PESCA1				PESCA2				PESCA3			
	Min	Max	M	SD	Min	Max	M	SD	Min	Max	M	SD
Age (years)	2.82	13.49	6.53	2.54	3.82	14.35	7.49	2.50	5.43	15.84	9.08	2.48
Weight (kg)	10.90	71.70	23.88	9.79	12.10	80.40	27.08	10.62	14.40	92.90	33.42	12.02
Height (m)	0.86	1.66	1.19	0.17	0.96	1.72	1.24	0.16	1.06	1.81	1.35	0.16
BMI (kg/m^2^)	11.92	29.61	16.22	2.49	12.85	31.80	16.92	2.68	12.82	32.53	17.73	2.99
**Age Group:**		**n**	**%**			**n**	**%**			**n**	**%**	
Under 6		94	45.40			75	36.20			17	8.20	
6 to 9.99		92	44.40			95	45.90			118	57.00	
10 and older		21	10.10			37	17.90			72	34.80	
Total		207	100.00			207	100.00			207	100.00	

M: mean; SD: standard deviation; Min: minimum; Max: maximum.

**Table 2 nutrients-16-03993-t002:** Overweight, PA/sports practice and healthy body composition prevalence evolution. Descriptive statistics.

		PESCA1				PESCA2				PESCA3				
		n	%	95% CI		n	%	95% CI		n	%	95% CI		*p*
Female	Overweight	16	17.39	9.65	25.14	15	16.30	8.76	23.85	21	22.83	14.25	31.40	
	Total	92	100.00			92	100.00			92	100.00			
Male	Overweight	21	18.26	11.20	25.32	26	22.61	14.96	30.25	34	29.57	21.22	37.91	**<0.001**
	Total	115	100.00			115	100.00			115	100.00			
**Total**	**Overweight**	**37**	**17.87**	**12.66**	**23.09**	**41**	**19.81**	**14.38**	**25.24**	**55**	**26.57**	**20.55**	**32.59**	**<0.001**
	**Total**	**207**	**100.00**			**207**	**100.00**			**207**	**100.00**			
Under 6	Overweight	10	10.64	4.41	16.87	8	8.51	2.87	14.15	21	22.34	13.92	30.76	**<0.001**
	Total	94	100.00			94	100.00			94	100.00			
6 to 9.99	Overweight	23	25.00	16.15	33.85	30	32.61	23.03	42.19	30	32.61	23.03	42.19	**0.023**
	Total	92	100.00			92	100.00			92	100.00			
Over 10	Overweight	4	19.05	2.25	35.84	3	14.29	0.00	29.25	4	19.05	2.25	35.84	
	Total	21	100.00			21	100.00			21	100.00			
Female	PA or sports practice prevalence	74	80.43	72.33	88.54	82	91.11	85.23	96.99	64	71.91	62.57	81.25	**0.007**
	Total	92	100.00			90	100.00			89	100.00			
Male	PA or sports practice prevalence	92	80.00	72.69	87.31	105	91.30	86.15	96.45	95	84.82	78.18	91.47	**0.015**
	Total	115	100.00			115	100.00			112	100.00			
**Total**	**PA or sports practice prevalence**	**166**	**80.19**	**74.76**	**85.62**	**187**	**91.22**	**87.35**	**95.09**	**159**	**79.10**	**73.48**	**84.73**	**0.001**
	**Total**	**207**	**100.00**			**205**	**100.00**			**201**	**100.00**			
Under 6	PA or sports practice prevalence	71	75.53	66.84	84.22	83	89.25	82.95	95.54	73	79.35	71.08	87.62	**0.018**
	Total	94	100.00			93	100.00			92	100.00			
6 to 9.99	PA or sports practice prevalence	80	86.96	80.07	93.84	86	93.48	88.43	98.52	70	78.65	70.14	87.16	**0.005**
	Total	92	100.00			92	100.00			89	100.00			
Over 10	PA or sports practice prevalence	15	71.43	52.11	90.75	18	90.00	76.85	100.00	16	80.00	62.47	97.53	0.417
	Total	21	100.00			20	100.00			20	100.00			
**Total**	**Healthy Body Composition**	**48**	**56.47**	**45.93**	**67.01**	**72**	**63.16**	**54.30**	**72.01**	**74**	**52.48**	**44.24**	**60.73**	0.103
	**Total**	**85**	**100.00**			**114**	**100.00**			**141**	**100.00**			

**Table 3 nutrients-16-03993-t003:** Daily screentime and rest time evolution. Descriptive statistics.

	PESCA1			PESCA2			PESCA3			
	M	SD	n	M	SD	n	M	SD	n	*p*
**Daily Screen time (h)**										
Female	1.53	0.78	82	1.55	0.65	82	2.06	1.12	82	**<0.001**
Male	1.67	0.77	108	1.81	0.88	108	2.47	1.31	108	**<0.001**
**Total**	**1.61**	**0.78**	**190**	**1.70**	**0.80**	**190**	**2.29**	**1.24**	**190**	**<0.001**
Under 6	1.51	0.76	88	1.55	0.83	88	1.81	0.93	88	**0.001**
6 to 9.99	1.59	0.75	84	1.72	0.68	84	2.66	1.35	84	**<0.001**
Over 10	2.18	0.76	18	2.33	0.83	18	2.96	1.23	18	**0.016**
**Daily rest time (h)**										
Female	10.16	0.69	88	10.30	0.52	88	10.08	0.63	88	**0.017**
Male	10.17	0.55	110	10.09	0.55	110	9.86	0.58	110	**<0.001**
**Total**	**10.17**	**0.61**	**198**	**10.18**	**0.55**	**198**	**9.96**	**0.61**	**198**	**<0.001**
Under 6	10.16	0.76	91	10.39	0.45	91	10.26	0.48	91	**0.019**
6 to 9.99	10.27	0.41	87	10.14	0.48	87	9.79	0.54	87	**<0.001**
Over 10	9.77	0.44	20	9.46	0.58	20	9.31	0.68	20	**0.003**

## Data Availability

The data presented in this study are available on request from the corresponding author. The data are not publicly available due to privacy and ethical restrictions, as they include sensitive information derived from medical records and clinical examinations.

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
