# Peer review of "Overweight Prevalence Changes Before and After COVID-19 in Spain: The PESCA Program Longitudinal Outcomes 2018–2021"

_nutrients, 2024, doi:10.3390/nu16233993_

Round 1
Reviewer 1 Report (Previous Reviewer 2)
Comments and Suggestions for Authors
STRUCTURE
- The manuscript is correctly structured. But there are some points to consider:
o Table 1: including the meanings of some abbreviations, such as M and SD, Min, Max. Table does not follow the magazine format also. Applicable to the rest of the document.
o Line 205: “we categorized body…” It is important to speak in the impersonal and not to speak in the first-person plural. Applicable to the rest of the document.
TITLE AND ABSTRACT
- The title mentions the type of research that has been carried out.
- The summary is well structured, listing all parts.
- Line 88: “Our contribution in this study,…”. Remember not to speak in the first-person plural.
INTRODUCTION
- In terms of scientific background, the current situation and prevalence of childhood obesity and its relation to physical inactivity is adequately mentioned.
However, below are some additional points that would help to better contextualise the study, to clearly define the scope and relevance of the PESCA programme, and to explain the effects of COVID-19 on child and adolescent overweight, providing a solid basis for research:
o 1. Detailed description of the PESCA methodology
Include specific information on how the programme is implemented: frequency of interventions, type of physical activities and nutrition education provided, and how these components are evaluated in participants.
o 2. Context of the PESCA programme in previous studies
Describe in detail whether there are previous studies or interventions similar to PESCA, both in Spain and in other countries, and their results on the prevention of childhood overweight. This would help to understand whether PESCA is a unique programme or whether it follows successful models from elsewhere.
Compare the methodological approaches of PESCA with other interventions to better position its relevance and replication potential.
o 3. Scientific rationale for the relationship between COVID-19 and overweight
Add evidence from studies linking decreased physical activity, increased screen time, and changes in sleep patterns with increases in BMI and risk of obesity in children.
o 4. Socio-demographic information on the sample
Indicate whether socio-demographic factors (age, gender, socio-economic status, geographic location) are considered in the analysis, as they may influence the effects of the pandemic and the PESCA programme on children's weight and habits.
This would allow analysis of whether COVID-19 restrictions will affect socio-economic or geographic groups differently.
o 5. Specification of the longitudinal analysis
Include a more detailed description of how the longitudinal design will allow assessment of differential effects before and after the pandemic.
Explain whether the analysis will assess time trends and whether longitudinal statistical models (eg, mixed-effects models) will be applied to capture individual-level changes.
o 6. Justification of study relevance
Explain the relevance of the study in terms of public health policy in Spain and the importance of monitoring and evaluating prevention programmes such as PESCA. That is, add how the results could influence future intervention strategies and public health decision making.
o 7. Expected impact of the study
Briefly outline how the findings could contribute to the design of future interventions tailored to emergency situations such as the pandemic.
- Regarding the specific objectives or hypotheses, it seems that the objectives are well defined. However, the study hypothesis should be added at the end of the abstract.
MATERIAL AND METHODS
- Study design. The research presents the key elements of the study design at the beginning of this section.
- In addition, if it describes the setting, location, relevant dates, the recruitment, exposure, follow-up and data collection periods, then the study design should be described.
- Eligibility criteria are stated, but sources and methods of participant selection are not. Incorporate the latter.
- Line 95: ‘Questionnaire: Clinical, personal, and family...’: What type of questionnaire is used? Has it been previously validated for this population? It would be interesting to provide some reference of this questionnaire in this type of population.
- Line 135: 2.1. . Subjects: Remove the dot in front of subjects.
RESULTS
- Overall, the results section is quite comprehensive and well-structured, addressing the different variables assessed (prevalence of overweight, body composition, physical activity, rest time and screen time) and presenting data segmented by gender and age group, as well as comparisons between the three measurement points (before and after COVID-19). However, to make it more complete and relevant, you might consider adding or strengthening some additional points:
o Interpretation of the Magnitude of Change:
Although statistical significance values and effect sizes are presented, it might be useful to include a brief analysis of the clinical or practical magnitude of these changes to understand whether the observed variations have important implications for participants' health. This is especially true in the context of public health interventions.
o Relationship between variables:
Consider analysing whether there is any variation between variables, for example, whether increased screen time is associated with higher BMI or lower physical activity. This could provide a more holistic view of how changes in one behaviour influence others and provide more comprehensive recommendations for intervention.
o Including these additional elements would enrich the results and facilitate the practical interpretation of the findings, as well as better contextualise the changes in a public health framework.
DISCUSSION
- To strengthen the discussion, some additional aspects could be addressed:
o International comparison: Broaden the discussion on how the effects observed in the PESCA sample are consistent with findings in other countries. This could include studies in different cultural and economic contexts, which would allow for an understanding of whether the impact on child and adolescent health is a universal phenomenon or whether there are important variations according to context.
o Analysis of socio-economic and regional factors: Include a reflection on how factors such as socio-economic status and access to resources (spaces for physical activity, technology for online classes, etc.) may have influenced the observed changes in child health during the pandemic. This could help contextualise the results and recognise possible inequalities that exacerbated the effects of confinement in certain populations.
o Psychological impact and relationship to physical health: Further explore the relationship between mental health and physical health habits, such as physical activity and screen time. Numerous studies suggest that the social isolation and stress of the pandemic have influenced the emotional health of children, and this may have an important connection to reduced physical activity and increased sedentary lifestyles.
o Long-term perspective and need for preventive programmes: Addressing the possible long-term consequences of increased overweight and sedentary lifestyles, considering that they tend to be associated with increased overweight and sedentary lifestyles.
o Limitations of subgroup analysis (age and sex): Although detailed analyses by age and sex are presented, it would be relevant to point out possible limitations of these subgroup analyses due to sample size or variability in adherence to the PESCA programme as a function of confinement, as these factors may affect the robustness of the comparisons.
CONCLUSION
- The conclusions section does not appear, it would be pertinent to rename some conclusions and limitations briefly and in line with aspects of improvement for future research.
Author Response
Please see the attachment.

Reviewer 2 Report (Previous Reviewer 3)
Comments and Suggestions for Authors
The manuscript's improvements largely align with my previous comments, particularly in terms of clarity and detail. I have no further request for this version.
Author Response
Thank you for your positive feedback. We are pleased that our revisions and improvements align with your expectations, and we appreciate your thoughtful review of our work.
Reviewer 3 Report (New Reviewer)
Comments and Suggestions for Authors
This study provides a valuable longitudinal perspective, enriching our understanding of how external factors, such as the pandemic, can disrupt lifestyle interventions in school settings. While the manuscript presents promising findings, clarity and additional context in several areas could enhance its overall quality.
1. The methodology is generally well-described, with appropriate statistical tests (Cochran's Q test and repeated measures ANOVA) selected for analyzing longitudinal data. However, while sample size and age distribution are mentioned, including additional demographic information, such as socioeconomic status, would provide more context about the sample. Moreover, more details about the school selection criteria would help assess the generalizability of the findings. A more thorough description of the specific components of the PESCA program, including the frequency of physical activity sessions and types of dietary education, would strengthen the reader's understanding of how each aspect may have contributed to the observed effects.
2. The discussion surrounding the observed stability of overweight prevalence before COVID-19 could be strengthened. Was this outcome expected? Were these changes significant when compared to trends in non-PESCA groups or national averages?
3. Comparing these findings with similar studies from other countries or regions would enhance the discussion. This would give readers a broader context regarding the impact of COVID-19 on youth overweight prevalence and lifestyle behaviors around the globe. Although the manuscript mentions the effects of lockdowns, it should also discuss other potential limitations, such as the influence of socioeconomic factors or changes in home environments during lockdowns. Acknowledging these factors could provide a more balanced perspective on the changes in overweight prevalence and lifestyle behaviors.
4. While the conclusions highlight the importance of intervention programs, there is limited discussion of the necessary adaptations for school-based programs following the pandemic. Recommendations for how PESCA or similar interventions could be adjusted to enhance resilience in the face of potential future disruptions would strengthen the study's practical implications.
5. Although the references cited in the manuscript likely include relevant sources on childhood overweight, the PESCA program, and the impact of COVID-19, the introduction would benefit from additional citations. Including more recent studies on the prevalence of childhood obesity, the effects of school-based interventions, and the pandemic's impact on youth health behaviors would enhance the manuscript's credibility and contextual relevance. I recommend mentioning the following references: Nally, S. et al., Children 2021, 8(6), 489; https://doi.org/10.3390/children8060489; Min-Ji, K. et al., Ann Pediatr Endocrinol Metab. 2024 Jun; 29(3):174-181. doi: 10.6065/apem.2346094.047. Epub 2024 Jan 29; Sares-Jäske, L. et al., Prev Med. 2022 Jul; 160:107095. doi 10.1016/j.ypmed.2022.107095. Epub 2022 May 17. PMID: 35594926.
6. There are instances of lengthy sentences containing multiple ideas. Breaking these into shorter, more concise sentences can improve readability and help maintain the reader's focus. For example, consider revising sentences that convey several points into separate statements.
7. While the manuscript is generally free from significant grammatical errors, thorough proofreading would be beneficial to catch any minor mistakes, such as misplaced commas or awkward phrasing. Pay attention to subject-verb agreement and ensure that all sentences are complete and well-structured.
Comments on the Quality of English LanguageOverall, the manuscript effectively conveys the essential findings and concepts. However, there are instances of lengthy sentences that contain multiple ideas. Breaking these into shorter, more concise sentences can improve readability and help maintain the reader's focus.
Author Response
Please see the attachment.

Reviewer 4 Report (New Reviewer)
Comments and Suggestions for Authors
Even with the publication of the study protocol, which details the research assumptions and strategies, I recommend a brief presentation of the design. Sometimes it appears to be a trial, and other times a follow-up cohort.
Therefore, it is also essential to present a reduced version of the intervention protocol.
I recommend using the CONSORT checklist to support the report.
Usually, intervention studies present “effect sizes”, which represent calculations between the initial and final means of the follow-up. It would be interesting to present these estimates.
Round 2
Reviewer 1 Report (Previous Reviewer 2)
Comments and Suggestions for Authors
No further comments.
This manuscript is a resubmission of an earlier submission. The following is a list of the peer review reports and author responses from that submission.
Round 1
Reviewer 1 Report
Comments and Suggestions for Authors
The study idea is smart and timely.
How did you define the sample size and the age range; actually, child psychophysiology and adolescent physiology are not comparable. This holds particularly true, as adolescents, by nature, do start to detach from parents’ supervision, including sleep, eating and leisure time behavior.
Abstract: “….in children attending PESCA program…”, does it mean that there were also control conditions? If so, how were participants assigned to the specific study conditions?
“…. prevalence significantly decreased..”; descriptively or statistically significantly decreased?
Please describe the type, duration, frequency, time and intensity of the intervention/program. Conclusions: from the information provided so far, the conclusions appear overstated.
Overall, the Abstract needs a thorough revision; specifically, the reader needs to know the key characteristics of the intervention, the rationale to assess a small sample size of 207 participants with a clearly very heterogenous psychosocial and psychophysiological set-up, including a relative increasing autonomy regarding the use of electronic devices, including SNSs, food intake and sleep behavior.
Introduction:
“As numerous globally published studies confirm, the Covid-19 pandemic has undeniably exerted a detrimental impact on the health of children and adolescents [5,6,7].”; actually, this is not true, and the authors are advised not to copycatting what tabloids wanted to sell us as truth. Please consider (Aghababa et al., 2021; Albrecht et al., 2022; Asmundson & Taylor, 2021) and revise your statements.
“The role of physical activity (PA) appears to be crucial, with PA time predominantly 59
decreasing due to the implementation of lockdowns [6,7,8].”; again, see (Aghababa et al., 2021) for a thorough and comprehensive overview of the state-of-the-art (at least till September 2021).
“…. for an extended period.”; please, be much more specific; this is crucial, as the time lapse of 42 days (March 15 to April 26) appears intuitively not sufficient to dramatically impact on a person’s behavior, in general, and on children and adolescents, in specific.
“…more substantial increases in BMI…”; please report the statistics, including effect sizes.
Please help the reader to understand the key components of the PESCA study.
Please formulate hypotheses and describe, whether and to what extent the present results may add to the current literature in a novel fashion.
Methods: Did you already publish some data?
Which rationale did you follow to assess these biophysiological information?
2.1 Subjects, better: Participants; if I got it right, the attrition rate was 85.21%, yes? if so, what might be the consequences of the present data?
Statistics: For F-tests, the correct effect size is partial eta-squared (Cohen, 1988, 1992; Cohen, 1994). For pairwise mean comparisons, please reference (Becker, 1988).
Results: As for now, the reader has difficulties to understand the big picture. Further, in my opinion, age should be introduced as covariate; the reason is: Figure 1; it appears plausible that participants crossing into adolescence are more prone to use electronic devices; please see (Hisler et al., 2020; Twenge et al., 2020; Twenge & Campbell, 2019; Twenge et al., 2017; Twenge & Martin, 2020; Twenge et al., 2018).
Why was Table 3 labelled as ‘qualitative’ when quantitative data are reported?
Figure 2, instead of PESCA as labels, consider that the reader would benefit from indications such as age, time lapse or similar.
References
Aghababa, A., Zamani Sani, S. H., Rohani, H., Nabilpour, M., Badicu, G., Fathirezaie, Z., & Brand, S. (2021). No Evidence of Systematic Change of Physical Activity Patterns Before and During the Covid-19 Pandemic and Related Mood States Among Iranian Adults Attending Team Sports Activities. Frontiers in psychology, 12, 641895. https://doi.org/10.3389/fpsyg.2021.641895
Albrecht, J. N., Werner, H., Rieger, N., Widmer, N., Janisch, D., Huber, R., & Jenni, O. G. (2022). Association Between Homeschooling and Adolescent Sleep Duration and Health During COVID-19 Pandemic High School Closures. JAMA Network Open, 5(1), e2142100-e2142100. https://doi.org/10.1001/jamanetworkopen.2021.42100
Asmundson, G. J. G., & Taylor, S. (2021). Garbage in, garbage out: The tenuous state of research on PTSD in the context of the COVID-19 pandemic and infodemic. J Anxiety Disord, 78, 102368. https://doi.org/10.1016/j.janxdis.2021.102368
Becker, B. J. (1988). Synthesizing standardized mean-change measures. British Journal of Mathematical and Statistical Psychology, 41(2), 257-278. https://doi.org/10.1111/j.2044-8317.1988.tb00901.x
Cohen, J. (1988). Statistical power analysis for the behavioral sciences (2nd ed.). Lawrence Erlbaum Associates.
Cohen, J. (1992). A power primer. Psychol Bull, 112(1), 155-159.
Cohen, J. (1994). The earth is round (p < .05). American Psychologist, 49(12), 997-1003. https://doi.org/10.1037/0003-066X.49.12.997
Hisler, G., Twenge, J. M., & Krizan, Z. (2020). Associations between screen time and short sleep duration among adolescents varies by media type: evidence from a cohort study. Sleep medicine, 66, 92-102. https://doi.org/10.1016/j.sleep.2019.08.007
Twenge, J. M., Blake, A. B., Haidt, J., & Campbell, W. K. (2020). Commentary: Screens, Teens, and Psychological Well-Being: Evidence From Three Time-Use-Diary Studies. Frontiers in psychology, 11, 181. https://doi.org/10.3389/fpsyg.2020.00181
Twenge, J. M., & Campbell, W. K. (2019). Media Use Is Linked to Lower Psychological Well-Being: Evidence from Three Datasets. The Psychiatric quarterly, 90(2), 311-331. https://doi.org/10.1007/s11126-019-09630-7
Twenge, J. M., Krizan, Z., & Hisler, G. (2017). Decreases in self-reported sleep duration among U.S. adolescents 2009-2015 and association with new media screen time. Sleep medicine, 39, 47-53. https://doi.org/10.1016/j.sleep.2017.08.013
Twenge, J. M., & Martin, G. N. (2020). Gender differences in associations between digital media use and psychological well-being: Evidence from three large datasets. Journal of adolescence, 79, 91-102. https://doi.org/10.1016/j.adolescence.2019.12.018
Twenge, J. M., Martin, G. N., & Campbell, W. K. (2018). Decreases in psychological well-being among American adolescents after 2012 and links to screen time during the rise of smartphone technology. Emotion (Washington, D.C.), 18(6), 765-780. https://doi.org/10.1037/emo0000403
Comments on the Quality of English Language
no comments
Reviewer 2 Report
Comments and Suggestions for Authors
he article titled "Overweight prevalence changes before and after Covid-19 in Spain. The PESCA program longitudinal outcomes 2018-2021" presents a longitudinal study on overweight prevalence in Spain before and after the COVID-19 pandemic using the PESCA program. The study examines the program's effectiveness in influencing health-related behaviors.
-
Objectives and Context: The study addresses a critical issue, given the increasing concern over overweight and obesity, especially in the context of the COVID-19 pandemic. Choosing the PESCA program as a focus is appropriate, considering its potential to impact health behaviors.
-
Methodology: The methodology is robust, including a longitudinal analysis with measurement points before and after the pandemic. The inclusion of 207 children and adolescents provides a significant sample for the study. However, it is recommended that the authors consider the effect of potential confounding variables in their analyses.
-
Results: The results indicate a significant decrease in overweight prevalence among PESCA program participants before the pandemic, which was reversed after the pandemic began. This finding emphasizes the negative impact of pandemic restrictions on healthy behaviors.
-
Discussion: The discussion appropriately integrates the results with existing literature, highlighting the role of physical activity and changes in screen time during the pandemic. The authors are advised to expand this section to include a more detailed discussion on the policy and practical implications of their findings.
-
Conclusion: The conclusions are consistent with the results presented, emphasizing the PESCA program's effectiveness before COVID-19 and the pandemic's negative impact on overweight prevalence. The authors are advised to highlight the need for adaptive public health strategies to mitigate the effects of future health crises.
-
Limitations and Future Directions: While the study acknowledges some limitations, it would be beneficial for the authors to discuss methodological limitations more thoroughly and suggest specific directions for future research.
- Next recommendations aim to enhance the article's contribution to understanding the impacts of health promotion programs and the pandemic on overweight prevalence among youth in Spain. Implementing these suggestions could provide deeper insights into effective strategies for promoting healthy lifestyles during and after public health crises:
-
Methodological Refinement: Authors should explore the inclusion of a more diverse sample to ensure the findings are generalizable across different demographics within Spain. Additionally, employing a control group not participating in the PESCA program could provide a stronger comparative analysis of the program's effectiveness.
-
Data Analysis: Introduce advanced statistical methods to analyze the interaction effects between participation in the PESCA program and pandemic-related lifestyle changes. This could involve regression models that account for interaction terms or path analysis to explore the direct and indirect effects of various factors on overweight prevalence.
-
Program Components Analysis: Conduct a detailed analysis of which specific components of the PESCA program were most effective in reducing overweight prevalence. This could involve subgroup analyses or qualitative feedback from participants to identify the most impactful aspects of the program.
-
Policy Recommendations: Based on the findings, develop specific policy recommendations for schools, communities, and health authorities to implement programs similar to PESCA. This could include guidelines on physical activity, nutritional education, and strategies to reduce screen time, particularly during periods of increased home confinement.
-
Longitudinal Impact Study: Recommend conducting follow-up studies to assess the long-term impact of the PESCA program on participants' health and well-being. This would involve tracking changes in overweight prevalence, physical activity levels, and other health indicators beyond the pandemic.
Reviewer 3 Report
Comments and Suggestions for Authors
This article examines the outcomes of a longitudinal program (PESCA) aimed at addressing pediatric overweight/obesity, both before and after the COVID-19 pandemic in Spain. The authors report that the program led to a decrease in some adverse outcomes. However, the positive effects were found to be offset by the impact of COVID-19 lockdown policies. This is a critical issue that warrants further investigation. I have some comments as follows:
1. The definition of overweight and obesity should be provided in the methods section, particularly for different age groups. While some information may be outlined in previously published protocols, given that the original version is in Spanish, some readers may not have access to it. Therefore, it is crucial to include important information in the text for clarity and accessibility.
2. The setting of the studied schools should be briefly described, specifically the Madrid and Castilla la Mancha Schools. In Table 1, the age of the included children ranged from 2 to 13 years old. Does this mean that the schools contained elementary school or kindergarten students?
3. In Table 1, the abbreviations should be listed.
4. 2.1 Subjects: In my opinion, this section is part of the results, because body weight and height are components of the outcomes measured.
5. L 190-191: The figures and tables need to be labeled in the order of their first appearance. For example, in Section 3.1, the overweight prevalence correlates with Figure 2, while the healthy fat section correlates with another table. Additionally, figures and tables should be positioned near the text where they are mentioned to facilitate understanding and contextualization. If this goal cannot be achieved, referring to figures and tables in the text as 'Figure 1' or 'Table 2' can serve as an alternative route.
6. L355-359: How to explain the consistent increase from stage 1 to stage 3? Encourage more discussion about this issue.
Comments on the Quality of English LanguageI have no comment on the quality of English writing.
Round 2
Reviewer 1 Report
Comments and Suggestions for Authors
"As pediatric professionals and healthcare personnel we are familiar with the physiological evolution of both physical and mental health in the sample. However, it is not the purpose of this paper to highlight these differences, but rather to describe the evolution of specific items related to cardiovascular health." I do believe this, though, this is not the point. The point is that compared apples with oranges, and while both apples and oranges belong to the group of 'fruits', this does not mean that we treat them equally. Similarly, the cognitive, emotional, behavioral and interactional patterns of children, who do basically depend from the family environment and parent behavior, are by no means comparable to adolescents' sociopsychological functioning. Given this, the authors must provide more and above all strong information and data that collapsing such psychophysiological completely different developmental age groups makes sense (see for instance also (Bethlehem et al., 2022; Keshavan et al., 2014; Lenroot and Giedd, 2006, 2011; Mills et al., 2014). One solution could be to run separate analyses for children and adolescents.
Dear Autors, please report verbatim the corrections made in the text. Information such as:
The decrease is statistically significant.
Line 34
are completely pointless, or do you really expect that Reviewers have that much time to compare the information provided in the point-by-point-response with the text?
"We do not question, of course, the data or conclusions of the study by Aghababa et al., 2021, referring to the adult population in Iran." I do not know exactly what you mean with this sentence. Should I get suspicious? Further, giving a closer look at this paper, the authors will find plenty of references arguing against the 'Covid-19-everything goes downhill'-attitude mainly circulated by the media.
"Cipolla et all.
Our data showed that patients whose BMI increased were more sedentary (p=0.024 for physical activity and p=0.005 for hours spent with videogames) during the pandemic.
There is no effect size in these work data." May be, but this is easy to calculate with the means and standard deviations (see also Becker 1988).
Most published studies providing anthropometric data, qualitative and quantitative data regarding physical activity in the pediatric population during the pandemic period, do so from a retrospective perspective, many of them based on surveys and/or forms or previous data records. The PESCA program began in 2018 with its main objective being overweight reduction. Our contribution in this study, compared to others, is to provide intra-subject longitudinal data. Obviously, we did not anticipate the pandemic when designing our program, and this allows for the analysis of real physical examination records and anthropometric and functional measurements taken before, during, and after the pandemic, enabling the evaluation of its impact on the selected sample." May be, though, this does not prevent the authors to formulate hypotheses and expectations, or did you run the study fully exploratory, without any predictions? - This seems rather impossible.
"Please verify that said information is already specified for the reader: in the case of age, in Table 1 and in the case of the date or time frame, in the text (lines 129-133). I'm sorry, but this is definitely not my job!
Becker, B.J., 1988. Synthesizing standardized mean-change measures. British Journal of Mathematical and Statistical Psychology 41(2), 257-278.
Bethlehem, R.A.I., Seidlitz, J., White, S.R., Vogel, J.W., Anderson, K.M., Adamson, C., Adler, S., Alexopoulos, G.S., Anagnostou, E., Areces-Gonzalez, A., Astle, D.E., Auyeung, B., Ayub, M., Bae, J., Ball, G., Baron-Cohen, S., Beare, R., Bedford, S.A., Benegal, V., Beyer, F., Blangero, J., Blesa Cábez, M., Boardman, J.P., Borzage, M., Bosch-Bayard, J.F., Bourke, N., Calhoun, V.D., Chakravarty, M.M., Chen, C., Chertavian, C., Chetelat, G., Chong, Y.S., Cole, J.H., Corvin, A., Costantino, M., Courchesne, E., Crivello, F., Cropley, V.L., Crosbie, J., Crossley, N., Delarue, M., Delorme, R., Desrivieres, S., Devenyi, G.A., Di Biase, M.A., Dolan, R., Donald, K.A., Donohoe, G., Dunlop, K., Edwards, A.D., Elison, J.T., Ellis, C.T., Elman, J.A., Eyler, L., Fair, D.A., Feczko, E., Fletcher, P.C., Fonagy, P., Franz, C.E., Galan-Garcia, L., Gholipour, A., Giedd, J., Gilmore, J.H., Glahn, D.C., Goodyer, I.M., Grant, P.E., Groenewold, N.A., Gunning, F.M., Gur, R.E., Gur, R.C., Hammill, C.F., Hansson, O., Hedden, T., Heinz, A., Henson, R.N., Heuer, K., Hoare, J., Holla, B., Holmes, A.J., Holt, R., Huang, H., Im, K., Ipser, J., Jack, C.R., Jr., Jackowski, A.P., Jia, T., Johnson, K.A., Jones, P.B., Jones, D.T., Kahn, R.S., Karlsson, H., Karlsson, L., Kawashima, R., Kelley, E.A., Kern, S., Kim, K.W., Kitzbichler, M.G., Kremen, W.S., Lalonde, F., Landeau, B., Lee, S., Lerch, J., Lewis, J.D., Li, J., Liao, W., Liston, C., Lombardo, M.V., Lv, J., Lynch, C., Mallard, T.T., Marcelis, M., Markello, R.D., Mathias, S.R., Mazoyer, B., McGuire, P., Meaney, M.J., Mechelli, A., Medic, N., Misic, B., Morgan, S.E., Mothersill, D., Nigg, J., Ong, M.Q.W., Ortinau, C., Ossenkoppele, R., Ouyang, M., Palaniyappan, L., Paly, L., Pan, P.M., Pantelis, C., Park, M.M., Paus, T., Pausova, Z., Paz-Linares, D., Pichet Binette, A., Pierce, K., Qian, X., Qiu, J., Qiu, A., Raznahan, A., Rittman, T., Rodrigue, A., Rollins, C.K., Romero-Garcia, R., Ronan, L., Rosenberg, M.D., Rowitch, D.H., Salum, G.A., Satterthwaite, T.D., Schaare, H.L., Schachar, R.J., Schultz, A.P., Schumann, G., Schöll, M., Sharp, D., Shinohara, R.T., Skoog, I., Smyser, C.D., Sperling, R.A., Stein, D.J., Stolicyn, A., Suckling, J., Sullivan, G., Taki, Y., Thyreau, B., Toro, R., Traut, N., Tsvetanov, K.A., Turk-Browne, N.B., Tuulari, J.J., Tzourio, C., Vachon-Presseau, É., Valdes-Sosa, M.J., Valdes-Sosa, P.A., Valk, S.L., van Amelsvoort, T., Vandekar, S.N., Vasung, L., Victoria, L.W., Villeneuve, S., Villringer, A., Vértes, P.E., Wagstyl, K., Wang, Y.S., Warfield, S.K., Warrier, V., Westman, E., Westwater, M.L., Whalley, H.C., Witte, A.V., Yang, N., Yeo, B., Yun, H., Zalesky, A., Zar, H.J., Zettergren, A., Zhou, J.H., Ziauddeen, H., Zugman, A., Zuo, X.N., Bullmore, E.T., Alexander-Bloch, A.F., 2022. Brain charts for the human lifespan. Nature 604(7906), 525-533.
Keshavan, M.S., Giedd, J., Lau, J.Y., Lewis, D.A., Paus, T., 2014. Changes in the adolescent brain and the pathophysiology of psychotic disorders. Lancet Psychiatry 1(7), 549-558.
Lenroot, R.K., Giedd, J.N., 2006. Brain development in children and adolescents: insights from anatomical magnetic resonance imaging. Neurosci Biobehav Rev 30(6), 718-729.
Lenroot, R.K., Giedd, J.N., 2011. Annual Research Review: Developmental considerations of gene by environment interactions. J Child Psychol Psychiatry 52(4), 429-441.
Mills, K.L., Goddings, A.L., Clasen, L.S., Giedd, J.N., Blakemore, S.J., 2014. The developmental mismatch in structural brain maturation during adolescence. Dev Neurosci 36(3-4), 147-160.
Comments on the Quality of English LanguageMinor English corrections required.
Reviewer 2 Report
Comments and Suggestions for Authors
No further comments
Reviewer 3 Report
Comments and Suggestions for Authors
The authors have adequately addressed the issues I mentioned. I have no further comment on this manuscript.